# Circulating Cell-Free Nucleic Acids: Main Characteristics and Clinical Application

**DOI:** 10.3390/ijms21186827

**Published:** 2020-09-17

**Authors:** Melinda Szilágyi, Ondrej Pös, Éva Márton, Gergely Buglyó, Beáta Soltész, Judit Keserű, András Penyige, Tomas Szemes, Bálint Nagy

**Affiliations:** 1Department of Human Genetics, Faculty of Medicine, University of Debrecen, H-4032 Debrecen, Hungary; szilagyi.melinda@med.unideb.hu (M.S.); marton.eva@med.unideb.hu (É.M.); buglyo.gergely@med.unideb.hu (G.B.); soltesz.beata@med.unideb.hu (B.S.); keseru.judit@med.unideb.hu (J.K.); penyige@med.unideb.hu (A.P.); 2Department of Molecular Biology, Faculty of Natural Sciences, Comenius University, SK-841 04 Bratislava, Slovakia; ondrejpos.sk@gmail.com (O.P.); tomas.szemes@geneton.sk (T.S.); 3Faculty of Pharmacology, University of Debrecen, H-4032 Debrecen, Hungary

**Keywords:** cell-free nucleic acids, nDNA, mtDNA, miRNA, lncRNA, circRNA, exosomes, biological fluids, liquid biopsy

## Abstract

Liquid biopsy recently became a very promising diagnostic method that has several advantages over conventional invasive methods. Liquid biopsy may serve as a source of several important biomarkers including cell-free nucleic acids (cf-NAs). Cf-DNA is widely used in prenatal testing in order to characterize fetal genetic disorders. Analysis of cf-DNA may provide information about the mutation profile of tumor cells, while cell-free non-coding RNAs are promising biomarker candidates in the diagnosis and prognosis of cancer. Many of these markers have the potential to help clinicians in therapy selection and in the follow-up of patients. Thus, cf-NA-based diagnostics represent a new path in personalized medicine. Although several reviews are available in the field, most of them focus on a limited number of cf-NA types. In this review, we give an overview about all known cf-NAs including cf-DNA, cf-mtDNA and cell-free non-coding RNA (miRNA, lncRNA, circRNA, piRNA, YRNA, and vtRNA) by discussing their biogenesis, biological function and potential as biomarker candidates in liquid biopsy. We also outline possible future directions in the field.

## 1. Introduction

Tissue genotyping is a traditionally accepted gold standard method in cancer diagnosis for identifying human genomic alterations. However, this technique has several limitations. It is a highly invasive, expensive and time-consuming method not applicable for small amounts of tumor tissue or in the follow-up of cancer treatments regimens. Liquid biopsy is a promising alternative for invasive diagnostic methods [1]. It is based on the detection of biomarkers present in body fluids. According to the National Cancer Institute, a biomarker is defined as “a biological molecule found in blood, other body fluids, or tissues that is a sign of a normal or abnormal process, or of a condition or disease”. An appropriate biomarker must be stable—in order to survive sampling and storing procedures—and easily detectable. Theoretically, all body fluids are suitable sources of biomarkers in liquid biopsies. The most frequently used one is the blood, but urine, cerebrospinal fluid, saliva, milk, seminal plasma, tears, etc., may also be used. The application of liquid biopsy has several advantages over tissue biopsies: the sampling procedure is easier, cheaper, repeatable, suitable for the monitoring of treatments and is not affected by the heterogeneity of tumors that is a main limitation of tissue biopsies in cancer [1].

Cell-free nucleic acids (cf-NAs) include several types of DNA and RNA molecules that are present in extracellular fluids (Figure 1). Cf-NAs originate from cells through cell death (such as necrosis or apoptosis) but active transport processes might also be involved in their transport [2,3,4]. The latter involves the formation of microvesicles (exosomes) or protein-cf-NA complexes, whereby cf-NAs show high stability in body fluids making them suitable biomarker candidates detectable by methods such as qPCR and sequencing. Cf-NAs are well-established biomarkers in prenatal diagnosis for the screening of genetic disorders in the fetus [5,6,7]. Accumulating evidence suggests that cf-NAs (cf-DNA and cell-free non-coding RNAs) might be promising biomarkers in the diagnosis and prognosis of cancer, cardiovascular or neurological diseases and diabetes. Here, we outline the different types of cf-NAs, their biological function and possible clinical application as well.

## 2. Short History of Cell-Free Nucleic Acids

Cell-free DNA (cf-DNA) was first detected in the sera of cancer patients in 1948, but the scientific community failed to recognize the importance of this finding for a long time [8]. Later, Tan et al. measured altered concentrations of cf-DNA in the blood samples of patients suffering from systemic lupus erythematosus [9]. The concentration of cf-DNA was estimated in the sera of healthy adults as well in 1975 [10]. Later, Leon et al. determined cf-DNA in malignant diseases and they suggested its use as a prognostic factor for determination of the efficacy of cancer treatment [11]. Sorenson et al. (1994) made a comprehensive study to compare extracted cf-DNA levels in healthy individuals and cancer patients [12]. RAS point mutations previously demonstrated in cancer tissue were present in cf-DNA samples as well [13]. At the time, the relationship was clearly established between the clinical status of an oncological disorder and the presence of certain cf-NAs in body fluids; still, no diagnostic method based on them had been introduced in the clinical practice yet. However, there was a growing interest in the field of prenatal diagnosis of genetic diseases as Lo et al. (1997) detected circulating fetal DNA in maternal plasma, showing the presence of Y-chromosome-specific DNA and RhD, thus laying the foundations of non-invasive prenatal genetic diagnostics [14]. The possibility to detect aneuploidies from cell-free fetal DNA extracted from maternal blood—which became possible after the introduction of next-generation sequencing (NGS) in 2011—attracted high levels of attention [15]. A new avenue was opened on the horizon of diagnostic procedures. It was possible to detect trisomies, subchromosomal aberrations and even monogenic disorders. Such success in prenatal testing encouraged clinicians and scientists working in oncology and other fields to continue their efforts towards developing reliable screening methods based on cf-NAs as biomarkers (Figure 1) [13].

## 3. Extracellular Vesicles

Homeostasis of the human body is based on and preserved by cell-to-cell interactions by using signaling molecules such as hormones, neurotransmitters and cytokines, sometimes involving extracellular vesicles (EVs). EVs were first described in 1946 [16]. They are classified by size, origin, content and function (Table 1). The main types of EVs are exosomes (30–100 nm), microvesicles (100–3000 nm), and apoptotic bodies (1000–5000 nm). Exosomes may originate from multivesicular bodies by exocytosis. Microvesicles are made by blebbing of the plasma membrane, while programmed cell death and apoptosis are the sources of apoptotic bodies. Accumulating evidence suggests that exosomes might have an essential role in intercellular communication. They are released by donor cells, transferred via body fluids and taken up by recipient cells making horizontal transfer of cf-NAs possible [5]. They may contain nuclear ssDNA, dsDNA, mtDNA, various RNAs, and even proteins and lipids. Cf-NAs can be detected not only in the lumen of exosomes but from their surface as well [17]. Analyzing the cargo of exosomes might lead to the discovery of potential biomarkers as well as a better understanding of the pathophysiology of donor cells.

## 4. Cell-Free DNA

### 4.1. Cell-Free Nuclear DNA (cf-nDNA)

Cf-DNA is present in body fluids in three different forms: free, attached to proteins (nucleosomes, Argonaute, HDL, LDL), or encapsulated in/attached to extracellular vesicles (exosomes, microvesicles, apoptotic bodies). The main characteristics of cf-DNAs are shown in Table 2. There are two main types: cell-free nuclear (cf-nDNA) and mitochondrial DNA (cf-mtDNA). The origin of cf-DNA is not fully explored. It is suggested that cf-DNA in the blood plasma of healthy individuals might originate from the apoptosis of lymphoid and myeloid cells [18,19]. In oncological patients, the major proportion of cf-DNA is released via apoptosis or necrosis of tumor cells [20]. In addition to cell death, neutrophils may mediate the immune response by releasing neutrophil extracellular traps (NETs) trapping and killing various pathogens [18]. These are extracellular network structures composed of both nuclear and mitochondrial DNA fibers, which are covered by various proteins such as histones and proteases [19]. High levels of NETs have been shown to correlate with levels of circulating DNA, suggesting that DNA released from cells during NETosis is involved in the formation of circulating DNA [21]. Cf-nDNAs are highly fragmented: 90% of total cf-DNA is present in low molecular weight [22]. Duque-Afonso et al. found that the majority of cf-DNA fragments range between 80 bp and 200 bp [23].

Clinical application of cf-DNA in non-invasive diagnostic tests has high potential (Table 2). Cf-DNAs are widely used in prenatal genetic testing (non-invasive prenatal testing, NIPT) in the detection of trisomies of autosomes and sex chromosomes. Microdeletions may also be detected with high sensitivity and specificity [24,25]. Since a large amount of cf-DNA may originate from tumor cells, identifying the source of cf-DNA may aid identification and localization of the disease. The fragmentation profile of cf-DNA may reveal tissue origin in cancer patients [26]. Furthermore, DNA methylation patterns are considered tissue-specific, so an analysis of plasma DNA methylation allows us to determine the origin of cf-DNA fragments [27]. Plasma cf-DNA methylation profiling has also been demonstrated as a very accurate approach for the diagnostics of several cancers [28,29]. Sequence analysis of tumor-associated cf-DNA might also be used in mutation identification. Several mutations have been identified in cf-DNA: e.g., *TP53*, *KRAS*, *BRAF*, *HER2* [30,31]. These might help clinicians in therapy selection. Furthermore, cf-DNA analysis might be applicable in staging and prognosis as well as monitoring response to therapy [32].

Based on their appearance in the circulation, cf-DNA molecules can be divided into the following basic classes: cell-free DNA fragments, vesicle-bound DNA and DNA-macromolecular complexes, which are described below.

### 4.2. Cell-Free DNA Fragments

Cf-DNA fragments are naked sequences that are not bound to any other molecules. During cell death, genomic DNA is cleaved and released into the circulation, but only DNA that is associated with proteins may resist cleavage by DNase, while free DNA fragments are completely lost in body fluids [22].

### 4.3. Vesicle-Bound DNA

Nucleic acids are found in EVs where they are protected from degradation [33]. Vesicles may be released from both healthy and tumor cells. EVs that are released from tumor cells contain various oncogenic factors in the form of oncoproteins and nucleic acids and are therefore named oncosomes. When an oncosome is introduced into a recipient cell, these factors may transfer the phenotype of tumor cells to idle cells [34].

### 4.4. DNA-Macromolecule Complexes

Circulating nucleosomes are DNA-protein complexes originating from the cleavage of chromosomal DNA during apoptotic cell death [35]. These are dsDNA fragments with the size of 180–200 bp that are wrapped around an octameric histone protein complex. Nucleosomes are linked to each other by 20–90 bp of DNA while the part wrapped around an octamer has an average length of 147 bp [18]. DNA bound directly to nucleosomes is protected from nuclease digestion, therefore in circulation it occurs in the form of mono- and oligo-nucleosome fragments [36]. Nucleosome spacing has been shown to vary between different cell types depending on tissue-specific gene expression. This is due to the fact that nucleosomes are located near the starting point of transcription, at the sites where transcription factors bind to DNA. These factors protect DNA from nuclease cleavage during apoptosis, resulting in a specific fragmentation pattern or transcription factor footprint, which provides information about the tissue of origin [22]. Felgner et al. designed DNA/cationic lipid complexes in a highly efficient transfection procedure. They showed that lipid particles spontaneously interact with DNA through the electrostatic interaction and facilitate the delivery of functional DNA into cells [37]. It was shown that lipids increase DNA half-life in the plasma of mice, suggesting that lipids protect DNA from degradation [38]. Thierry et al. propose that the formation of DNA-lipid complexes results from the universal tendency of nucleic acids to self-organize with cationic lipids and suggest possible existence of ubiquitous self-organization of genetic materials [39,40]. Virtosomes are DNA-RNA-lipoprotein complexes. Lipoproteins bound in virtosomes protect nucleic acids from degradation by nucleases. Nucleic acids, proteins and lipids involved in the formation of these complexes are actively released from living cells and it is supposed that they play a role in intercellular communication [18]. Virtosomes may enter recipient cells, incorporate their DNA into their genome and subsequently modify the biology of these cells. These modifications include various immunological changes or a transformation of normal cells into tumor cells.

### 4.5. Cell-Free mtDNA

Mitochondria are ubquitous eukaryotic cell organelles with α-proteobacterial origin, playing a central role in energy and biomolecule production, cell proliferation as well as apoptosis. The circular, bacterial-like mitochondrial genome is present in 100–10,000 copies and has a length of 16,569 bp in human cells [41]. Multiple forms of cf-mtDNA exist in body fluids, including exposed cf-mtDNA fragments, mtDNA contained within vesicles or microparticles, and extruded whole mitochondria [42]. Cf-mtDNA may originate from autophagy, apoptosis, necrosis or active transport (secretion) processes [43]. MtDNA may also be released during the formation of NETs [44]. Cf-mtDNA fragments are thought to be generally shorter than cf-nDNA [45], ranging from 30–80 bp with peaks at 42–60 bp, but longer fragments (220 bp) were also reported [43,46]. Like genomic DNA, mtDNA may also be transported by microvesicles [47]. It is interesting to note that microvesicles may harbor a full mitochondrial genome that may be transferred to cells with metabolic dysfunctions, restoring their metabolic activity. On the other hand, it has been shown that such horizontal transfer of mtDNA by microvesicles may awaken dormant tumor cells and induce resistance to chemotherapeutic agents [48,49]. It was also demonstrated that both normal cells and cultured tumor cells are able to secrete their intact, respiratory competent mitochondria [50]. Several studies have linked cf-mtDNA copy number changes to various human diseases, including cancer, neurodegenerative or cardiovascular diseases [51]. Reduced cf-mtDNA levels in the cerebrospinal fluid of neurodegenerative disease patients—including Alzheimer’s and Parkinson’s disease as well as multiple sclerosis—have also been reported. However, although reduced cf-mtDNA might be a hallmark of neurodegeneration, the utility of cf-mtDNA as a biomarker of such diseases is limited [52,53].

Apart from copy numbers, the fragmentation of cf-mtDNA may also represent a potential biomarker. Since DNA inside mitochondria is protected from nuclease cleavage, it remains relatively intact. Therefore, short cf-mtDNAs are more likely to have been released by tumor cells as they mainly undergo necrosis, while apoptosis is more common in normal cells. This hypothesis is supported by a study of An et al., who showed the size of blood cf-mtDNA fragments to be inversely correlated with tumor burden in cancer patients [54]. Apart from mtDNA fragments, the existence of full length mtDNA in cell-free human plasma has been reported. Furthermore, it was successfully used to perform haplogroup matching [55].

## 5. Cell-Free Microbial and Viral DNA

Cf-DNA may also originate from the human microbiome, which comprises a vast corpus of bacterial, archaeal, viral, and fungal microbial taxa. The predominant microbial cf-DNAs in plasma originate from bacteria [56]. Since several body fluids are in close contact with certain tissues, they may contain genetic material from local microbes. Human blood has been considered a sterile microenvironment in which bacteria appear only periodically, however, evidence for the existence of a microbiome in the blood of healthy individuals is accumulating [57]. Gosiewski et al. suggest that bacteria continuously translocate into the blood, but not always cause sepsis [58]. Since the composition of the microbiota modulates host physiology—including metabolism, inflammation, immunity, and hematopoiesis—it may play a role in tumorigenesis, cancer progression, and response to therapy [59]. Moreover, there is an association between diverse types of cancer and specific microbiota [60], suggesting that detection of microbiome cf-DNA in blood could provide an accurate and non-invasive testing method for cancer diagnosis [61] or information to improve prognosis and treatment in cancer patients [56].

Fragments of cf-DNA from pathogens causing infections at various locations in the body were also found in peripheral blood [62], so microbial cf-DNA analysis could represent a non-invasive approach to provide rapid, actionable treatment information for invasive fungal infections when a biopsy is not possible [63]. Witt et al. reported the detection of relevant microbial cf-DNA in maternal plasma of patients with chorioamnionitis. Their results may contribute to the development of a non-invasive assay for the detection of fetal exposure to microorganisms that would be useful to reduce complications from early neonatal sepsis [64]. Apart from blood, other body fluids may also contain microbial cf-DNA that may have clinical value. Urinary cf-DNA was demonstrated as a highly versatile marker to monitor bacterial and viral urinary tract infections. The assay has the potential to become a valuable tool to monitor bacteriuria and viruria in kidney transplant cohorts and to ascertain their potential impact on allograft health [65]. Since lymphatic fluid is in contact with neurons, it can be used for the detection of neuronal pathogens. On the other hand, testing vaginal discharge and semen samples may reveal pathogens associated with sexually transmitted diseases [66]. Several studies indicate detection of microbial cf-DNA by next-generation sequencing as an accurate approach to identify and quantify pathogens in clinical samples. However, in order to correctly evaluate such data, it is important to deepen our understanding of the involvement of microbes in the development of disease [66].

## 6. Cell-Free RNA

RNA molecules have also been detected in extracellular fluids (Figure 2). The origin of these molecules might be passive leakage from dead cells (e.g., linked to necrosis or apoptosis), or active secretion that might occur via membrane-bound vesicles (e.g., microvesicles) or follow a vesicle-free RNA-binding protein dependent pathway (e.g., with HDL, AGO2, NPM1 proteins) [2,67]. The latter mechanisms are thought to be well-regulated, energy-requiring processes. Several kinds of cell-free RNAs (cf-RNAs) have been detected [68]. Although coding mRNA was also observed in the circulation of oncological patients [69,70], most studies focus on cell-free non-coding RNAs (cf-ncRNAs) due to their relatively high stability resulting from their association with proteins or incorporation into vesicles that protect them from the degrading effects of RNases. Several studies suggest that the expression of cf-ncRNAs differs between disease patients and healthy individuals [71,72,73,74,75,76], suggesting them as suitable biomarker candidates for diagnostic tests. However, the biological significance of cf-RNAs is not well described yet. It is suggested that cf-RNAs might be involved in cell-to-cell communication and act as hormones [2,67]. These might affect tumor microenvironments contributing to tumor growth and invasion. Here we provide information about the most studied cf-ncRNAs regarding their biogenesis, biological function and their application as biomarkers.

### 6.1. MicroRNAs

The most studied ncRNA molecules are microRNAs (miRNA), which are small (18–23 nt) ssRNA molecules involved in posttranscriptional gene regulation. MiRNA biogenesis is a well-described process that starts with the transcription of the primary miRNA transcript, processed to mature miRNA by the enzymes Drosha and Dicer [77,78]. Mature miRNAs are incorporated into the RISC complex and target the 3′ UTR region of certain mRNAs, leading to mRNA degradation or the block of translation depending on the extent of complementarity [71,77,78]. More than 2500 miRNAs have been identified in the human genome to this date and they are involved in the regulation of several biological processes including development, cell cycle or apoptosis. Furthermore, one miRNA affects a number of target mRNAs and a single mRNA may contain target sites for multiple miRNAs. This results in mRNA–miRNA networks that may have a prominent role in posttranscriptional gene regulation and affect cellular physiology [79,80,81]. The significance of miRNAs in the development of diseases is confirmed by the observation that several disorders are accompanied by altered miRNA expression. This holds true for cf-miRNAs as well [71,72,73,81,82,83,84,85]. It is suggested that tumor-derived cf-miRNAs incorporated into vesicles are delivered to recipient cells, where they have an influence on gene expression [2,67]. This might affect the tumor microenvironment favoring tumor growth and invasion. Cf-miRNAs play a key role in the transition of normal fibroblasts to cancer associated forms by promoting cancer progression (e.g., by influencing the extracellular matrix or the secretion of growth factors) and affect the polarization of tumor associated macrophages that might promote tumor growth by affecting angiogenesis, matrix remodeling or suppressing inflammation [2]. However, the biological significance of cf-miRNAs might be much broader. They are involved in the crosstalk between mother and fetus during fetal development as blastocyst-secreted miRNAs are taken up by the endometrium, promoting implantation [86,87]. Moreover, since several immune-related miRNAs were identified in breast milk suggesting their role in the communication between mother and child, maternal cf-miRNAs might also influence fetal development even after birth [67]. It is also confirmed that cf-miRNAs are involved in host-parasite communication [67]. More studies are required to explore the versatile role of miRNAs and cf-miRNAs in cellular physiology. Understanding their significance might open new avenues in cancer therapy and diagnostics. Targeting miRNAs as an anticancer therapeutic strategy (by inhibition or replacement) has the potential to change cancer phenotype [2,88]. MiR-34a is considered a promising candidate for this purpose, as it has a tumor suppressor function and its expression differs in cancer patients compared to healthy individuals in several cancer types [73,89,90,91]. Its introduction to cancer cells decreased the rate of cell proliferation and invasion, bringing forth an increased rate of apoptosis [92,93]. Cf-miRNAs are considered to be useful cancer biomarker candidates on the basis of their altered expression in tumors. Their relatively high diagnostic value is supported by numerous studies and is extensively reviewed elsewhere [71,72,73,81,82,94]. It is important to note that the diagnostic potential of cf-miRNAs might be further increased by the application of multivariate diagnostic tests that combine more than one cf-miRNA or cf-miRNAs with other biomarkers [95,96]. The application of multivariate diagnostic tests possesses high promise in future cancer diagnostics.

### 6.2. Long Non-Coding RNAs

Long non-coding RNAs (lncRNAs) are non-protein-coding transcripts with a length of more than 200 nt. They can be transcribed from intergenic regions (long intervening non-coding RNAs), from the introns of protein-coding genes (intronic lncRNAs) or as antisense transcripts of genes [75]. They have broad molecular functions: they may be involved in the epigenetic regulation of allelic expression (e.g., Xist in X chromosome dosage compensation in female mammals), they may act as scaffolds for protein complexes or as decoys for specific target molecules to limit their availability (e.g., lncRNAs possess binding sites for miRNAs, regulating their abundance). They may also serve as precursors for small non-coding RNAs (sncRNA) or be involved in post-transcriptional gene regulation (e.g., antisense lncRNAs binding to their corresponding sense transcripts and alter splice-site recognition or spliceosome recruitment in mRNA processing) [75,97]. Their expression correlates well with many diseases. Several lncRNAs have been identified that might be involved in cancer development (e.g., MALAT1, HOTAIR, CCAT1) or in cardiovascular diseases (e.g., LIPCAR, ANRIL, KCNQ1OT1) [74,75]. Furthermore, lncRNAs are confirmed to have a tissue specific expression pattern that might contribute to the heterogeneity of tumors [98]. Many of these lncRNAs proved to have a detectable expression level in plasma samples, making them promising biomarker candidates in non-invasive diagnostics. Furthermore, their extracellular presence suggests a role in cell-to-cell communication akin to cf-miRNAs, however, limited studies have been done in this field yet. It was reported that cf-lncRNAs might promote angiogenesis by affecting endothelial cells [99]. It is also hypothesized that cf-lncRNAs might affect the miRNA secretome of tumor cells, alterin the responsiveness of surrounding cells to cf-miRNAs [98]. The competition of lncRNAs and mRNAs for a common pool of miRNAs suggests a complex regulatory network, sparking the competitive endogenous RNA (ceRNA) hypothesis aiming to explain the involvement of non-coding RNAs in cancer progression and migration [99,100,101]. The main limitation of this hypothesis is that it requires high concentrations of the involved RNAs, therefore, ceRNA networks are thought to depend on cellular and extracellular compartmentalization. Exosomes are considered particularly suitable for such interactions as many RNA components may reach high concentrations inside [99].

### 6.3. Circular RNAs

Circular RNAs (circRNA) are produced mostly from protein-coding genes by a process called backsplicing, in which a downstream splice-donor site is covalently linked to an upstream splice-acceptor site [102]. They consist of 1 to 5 exons and 0 to 4 internal introns. Although they used to be considered as “junk” molecules generated by aberrant splicing, nowadays thousands of circRNAs have been identified in many organisms and they are thought to have important biological functions. They were shown to act similarly to miRNA- and protein-sponges or scaffolds, but they might also act as templates for cap-independent translation using internal ribosome entry sites [102]. They proved to exhibit abundant, cell-type and tissue-specific expression. High expression was demonstrated in the brain, which is enriched in neuronal protein coding genes suggesting their potential role in the central nervous system [103]. The identification of a circular form of the sex determining region Y (SRY) transcript suggests its role in development [103]. Furthermore, several circRNAs were found to be associated with the development of cancer [76]. Due to their circular form, they are considered to have high stability that is also promoted by their appearance in exosomes in case of their cell-free counterparts, making them promising biomarker candidates for cancer diagnostics [76]. Cf-circRNA expression proved to be associated with cardiomyopathy and with colorectal, thyroid or endometrial cancer [104,105,106,107]. It is also hypothesized that exosomes containing circRNAs might be taken up by recipient cells with the potential to affect post-transcriptional regulation of gene expression, thus, they have influence on the tumor microenvironment [76]. However, more studies are required in this field in order to evaluate the role and potential of circRNAs.

### 6.4. PIWI-Interacting RNAs

PIWI (P-element Induced Wimpy testis in *Drosophila*)-interacting RNAs (piRNAs) are animal-specific small (24–32 nt) ncRNAs that bear 2′-O-methyl-modified 3′ termini and guide PIWI-clade Argonaute proteins (PIWI proteins) [108]. They are mostly processed from long single-stranded precursors that are transcribed from genomic loci called piRNA clusters. The best-known function of piRNAs is silencing transposons in the germ line, involving both transcriptional (by heterochromatin formation) and post-transcriptional (by mRNA cleveage) mechanisms. It was also discovered recently that aberrant expression of piRNAs is associated with cancer cell proliferation, differentiation, progression and metastasis formation in a number of cancer types, suggesting their role in tumor formation as either oncogenes or tumor suppressors [109]. However, fewer reports are available about cell-free piRNAs. They were detected in the serum of patients suffering from renal cell carcinoma [110] and were associated with DMD as well [111]. These results are not yet sufficient to make any meaningful conclusions about the biological or prognostic role of cf-piRNAs, so we are looking forward to more studies published in the field.

### 6.5. YRNA

YRNAs are relatively small, 100 ± 20 nt long non-coding molecules that fold into a characteristic stem-loop secondary structure. Four highly conserved YRNA genes are present in the human genome (RNY1, RNY3, RNY4 and RNY5) that are transcribed by RNA polymerase III from their individual promoters [112]. Although they were discovered in 1981, their exact biological function is poorly understood. They were detected as components of the ribonucleoprotein (RNP) complex with Ro60 in the samples of patients with autoimmune diseases. Studies with RoRNP concluded that RoRNP interacting YRNAs are involved in the regulation of RNA stability and cellular stress responses. They may also affect the initiation of DNA replication by interacting with several initiation proteins (e.g., Cdc6, Cdt1 and DUE-B) [112]. YRNA expression proved to be altered in prostate cancer and in clear cell renal cell carcinoma [113,114]. Small RNA fragments derived from YRNAs (YsRNAs) are present in apoptotic or proliferating cells as well as in the brain, retina and healthy mammalian tissues. YsRNA biogenesis seems to be independent of the canonical miRNA biogenesis pathway and YsRNAs do not have gene silencing activity, thus, little is known about their biological relevance. YsRNA expression proved to be elevated during apoptosis in macrophages and YsRNAs activate cell death and inflammation in cultured monocytes/macrophages [115]. YsRNAs have also been detected with transfer RNA (tRNA) fragments circulating in blood samples. These cf-YsRNAs are incorporated into vesicles or cell-free RNP complexes and are derived from the 5′ and 3′ termini of full-length YRNAs [112,116]. The level of cf-YsRNAs as well as 5′-tRNA-derived fragments was reported to be significantly different in patients with breast cancer or oral squamous cell carcinoma, but they were identified in the blood samples of healthy individuals as well [116,117,118]. The function of extracellular YsRNAs is not well understood, but most of them are thought to be degradation products of highly abundant cellular YRNAs. However, as the function of cellular YRNAs has not been well-described either, their role in intercellular communication cannot be excluded. Their clinical potential might also be high as diagnostic biomarkers [112].

### 6.6. Vault RNA

Vault RNAs (vtRNAs) are sncRNA molecules present in a broad spectrum of eukaryotes as a component of barrel-shaped RNP particles also known as vaults. Four human paralogs (vtRNA1-1, vtRNA1-2, vtRNA1-3, vtRNA2-1) with lengths of 88–100 nt have been described. Although their exact function remains unclear, these molecules have been associated with cellular processes such as drug resistance, apoptosis and nuclear transport [119]. In general, vtRNAs are upregulated in cancer cells that are resistant to several chemotherapeutic compounds. It was shown that a suppression of vtRNA expression in cancer cells led to the progressive decrease of resistance to mitoxantrone, suggesting its role in mitoxantrone resistance in malignant cells [120]. To our knowledge, there are no studies discussing extracellular vtRNA, however, since these molecules are bound to RNPs, we may assume their stability in extracellular space.

### 6.7. Other Non-Coding RNAs

Several other cellular non-coding molecules have been detected in body fluids as well [68]. Some of them have been shown to undergo processing into smaller RNA fragments, such as tRNA fragments created by cleavage of tRNA near the anticodon loop, also known as 5′ tRNA halves. These fragments are abundant in blood cells suggesting their hematopoietic and lymphoid origin in serum samples, where they are mostly present in nucleoprotein complexes [121]. Several mechanisms for the regulation of gene expression by such tRNA fragments have been described, including repression of translation by inhibiting the assembly of the translation initiation machinery, or a sequence-specific gene regulation. However, sufficient understanding of how these fragments regulate gene expression is still lacking [122]. As 5′ tRNA halves may regulate mRNA translation, Dhahbi et al. suggested their function as signaling molecules [121]. Other non-coding RNAs such as rRNA, small nuclear RNAs (snRNA) and small nucleolar RNAs (snoRNAs) have also been identified in human blood [68]. There might still be other types of RNA present in the circulation that could be studied and possibly harvested as biomarkers (Figure 2).

## 7. Cf-NA Detection

Detection and quantification of circulating cell-free nucleic acids is still challenging, which limits their use in medicine. Problems in determination are mostly due to their short length and low concentrations in body fluids. Nevertheless, several approaches have been developed to detect cell-free nucleic acids. Methods of cfDNA-based mutation profiling are centered around two main approaches: (i) PCR-based methods that target specific mutations in DNA where priori knowledge of DNA mutations is required, (ii) whole exome/genome sequencing (WES, WGS) based methods that allow the detection of all possible aberrations in DNA. To detect cancer-associated alleles in blood, a variety of new PCR methods were established besides the widely used real-time PCR (RT-PCR), such as allele-specific amplification PCR (AS-PCR), allele-specific non-extendable primer blocker PCR (AS-NEPB-PCR), peptide nucleic acid-locked nucleic acid (PNA-LNA) PCR clamp and co-amplification at lower denaturation temperature PCR (COLD-PCR). Digital PCR (dPCR) methods, including droplet digital PCR (ddPCR) and BEAMing (beads, emulsions, amplification and magnetics) have much higher sensitivity (Table 3) [1,22,32,123]. Next-Generation Sequencing (NGS) is also widely used in tumor tissue profiling; however, it has limited analytical sensitivity in cfDNA applications. This is typically due to the efficiency by which genetic regions of interest can be captured/enriched from cfDNA and the higher error rate of sequencing reactions. These limitations can be overcome by using either PCR amplification of the target regions using region-specific primers, or hybridization-based capture of target regions using complementary oligonucleotides followed by the amplification of the captured DNA library. The recently introduced CAPP-seq method has an improved sensitivity, it can use 7 ng (1100 genome equivalent) input ctDNA and its detection limit is 0.00025% mutant alleles (Table 3) [124,125,126].

Several methods exist for the detection of cell-free non-coding RNAs as well. RNA-Seq utilizes NGS for high-throughput analysis of cell-free RNA that is applicable for both RNA expression detection and novel RNA discovery and considered to be the most sensitive technique for miRNA, lncRNA and circRNA identification and quantification. However, it is time consuming, it has high cost per analysis and requires expertise due to specific sequencing library preparation protocols and bioinformatic analysis of data [127,128,129]. Using microarrays for nucleic acid detection and analysis is a well-established method used previously for multiplexed analysis of protein coding genes. Nowadays, the method is also used for rapid and high throughput detection and quantification of miRNAs and lncRNAs. Since it is based on hybridization methodology and does not involve PCR amplification of the sample, it has lower sensitivity than RNA-Seq, but microarrays are not prone to PCR-introduced errors and bias [130,131,132]. Recently, the NanoString nCounter System—a hybridization-based digital count technology—was also introduced. It is capable of highly multiplexed single molecule counting using molecular barcodes and fluorescent scanning technology to provide digital detection. The method does not involve an amplification step and measures nucleic acid content by counting molecules directly. However, it requires larger amounts of input cell-free nucleic acids for analysis and data normalization for miRNAs is still problematic. The platform can also detect copy number variations in 87 genes known to be amplified or deleted in cancer [133]. The method still considered as the “gold standard” for diagnostic purposes is the RT-qPCR. For detection of short RNA molecules (including miRNA), adaptions of the standard RT-qPCR techniques have been established [134]. The advantage of these methods is that they can be performed on relatively standard equipment, the tests are widely available and can be automated. Since the method has high sensitivity, specificity and reproducibility, it is still the most attractive method for routine testing in clinical laboratories.

## 8. Current Perspectives on the Role of cfNAs as Biomarkers

Cf-DNA is a widely applied biomarker in prenatal diagnosis for characterizing fetal genetic disorders. Increasing effort has been aimed recently at the utilization of cf-NAs as diagnostic biomarkers in other diseases, such as cancer [1]. The reason: while important advances have been made in recent years in the field of cancer biology, a reliable set of biomarkers to diagnose and monitor disease is often lacking [135]. Currently, the use of circulating cf-DNA and cf-RNA markers is investigated in cancer prediction and screening, monitoring treatment and recurrence, detection of resistance to therapy as well as minimal residual disease [136,137,138]. Despite their great promise in cancer diagnostics, only a few cf-NA-based liquid biopsy tests have been approved for clinical use. The reasons for this are the following: (i) an overall poor understanding of how components of intracellular RNA networks behave when released into the circulation, (ii) tumor-derived DNA is found in a low concentration and is masked by an—up to 1000-fold—excess of non-informative DNA originating from other cells, and (iii) a gap between theoretical and clinical oncologists resulting in some reports on novel biomarkers with little regard to practical aspects [135]. On the whole, overwhelming information is available about promising biomarker candidates, but few studies focus on the combination of these biomarkers as a recommended diagnostic method that represents the most likely future direction in the field. Still, despite all obstacles and limitations, if we take the future development of molecular diagnostics and bioinformatics into account, the application of liquid biopsy based multivariate diagnostic tests as a routine in healthcare must not be far. These tests will open a new path in personalized medicine in the future.

## Figures and Tables

**Figure 1 ijms-21-06827-f001:**
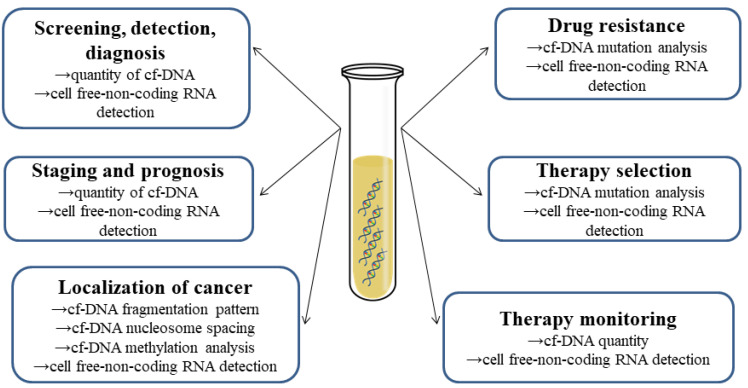
Utilization of cell-free Nucleic Acids in cancer diagnostics and treatment.

**Figure 2 ijms-21-06827-f002:**
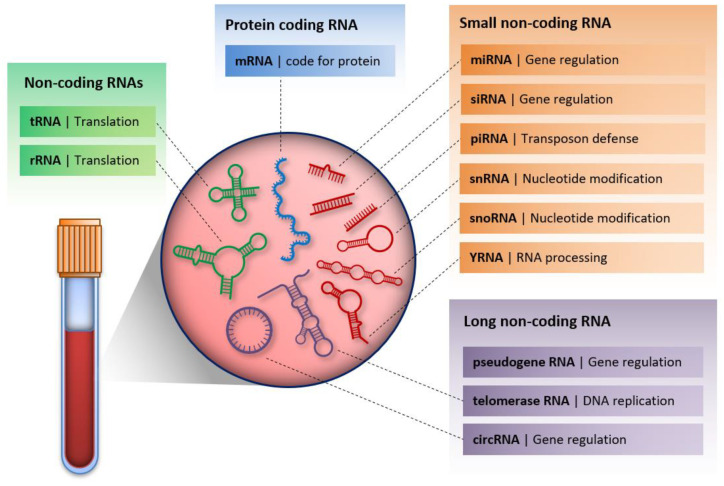
Cell-free RNA molecules in serum/plasma with their biological function.

**Table 1 ijms-21-06827-t001:** Main characteristics of extracellular vesicles.

Extracellular Vesicle Type	Size	Plasma/Serum Concentration	Origin	Content	Markers	Function	Morphology
Exosome	40–100 nm	5.3 particle/mL × 10^6^	Exocytosis from multivesicular bodies	Proteins, lipids, DNA, mRNA, miRNA, lncRNA, circRNA	Alix, Tsg101, tetraspanins (CD81, CD63, CD9), flotilin	Intracellular communication	Cup-shape
Microvesicle	50–3000 nm	5–50 µg/mL	Outward budding of plasma membrane	Proteins, lipids, mRNA, miRNA, ncRNAs	Phosphatidylserin, Integrins, selectins, CD40	Intracellular communication	Cup-shape
Apoptotic body	800–5000 nm	Much less compared to EVs and EXOs (Lazaro-Ibanez 2014)	Programmed cell death or apoptosis	Nuclear fractions, cell organelles, proteins, mRNA, ncRNA, DNA	Annexin V, phosphatidylserin	Facilitate phagocytosis	Heterogeneous

**Table 2 ijms-21-06827-t002:** Cell-free DNA in serum/plasma.

Full Name	Size	Concentration	Clinical Application
Genomic DNA	166->10.000 bp	13.9 ± 3.7 mg/L	Prenatal testing, diagnosis of cancer, mutation detection, cancer localization
Mitochondrial DNA	20–100 bp; <1–21 kbp	4.21 ± 0.38 copies/L	Diagnosis of cancer
Microbial DNA	Variable	20–450.000 microbe specific cfDNA/µL	Diagnosis of microbial infections and cancer

**Table 3 ijms-21-06827-t003:** PCR- and NGS-based method for mutation analysis of cfDNA. MAF: Minimum Allele Frequency.

Method	Platform	MAF	Specificity	Limitations
NGS	WGS/WES	0.02%	80–90%	High ctDNA input
	CAPP-Seq	0.00025%	>99.99%	High ctDNA input; detects only known mutations
Digital PCR	ddPCR	0.1%	100%	Detects only known mutations; limited in multiplexing
	BEAMing	0.01%	100%	Detects only known mutations
Real-Time PCR	qPCR	0.1%	99%	Detects only known mutations
	AS-PCR	1%	98%	Detects only known mutations
	PNA-LNA PCR clamp	0.1–1%	79%	Detects only known point mutations
	COLD-PCR	0.1%	94.9%	Detects limited genomic loci; limited in multiplexing

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
