# Peer review of "Circulating Cell-Free Nucleic Acids: Main Characteristics and Clinical Application"

_ijms, 2020, doi:10.3390/ijms21186827_

Round 1
Reviewer 1 Report
Dr. Szilagyi and colleagues provide an overview about all known cfNAs including cfDNA, cf-mtDNA and cf-non-coding RNA molecules by discussing their biogenesis, biological function and potential as biomarker candidates in liquid biopsy. Authors also discuss the future directions of these highly promising diagnostic tool.
Critique
The manuscript is concise, scientifically accurate, and interest to research scientist, pharmacologist, physician scientist and clinicians. There is no ethical concerns, including similarity between the reviewed manuscript and any published article. There minor problems (see below) that can be easily corrected.
Minor:
--line 14: “CfDNA …. CfNA???
--line 15: Delete “In cancer ……”
--line 38: “ …..in liquid biopsies (Figure 1.). Delete period within parentheses.
--line 39: “… liquor ….” Commonly alcoholic fluid please change. Please use cerebrospinal fluid (as an example).
--lines 45-46: “……… also be involved in their transport [2; 3; 4] Delete: Jahr et 46 al 2001).
--lines 60-61 The sentence will better sound “Sorenson et. al., (1994), made a comprehensive study to determine the isolated cfDNA from healthy individuals and cancer patients [12].
--line 62: “RAS” spell it out.
--line 66: “…… Lo et al. (1997) detected circulating fetal DNA (cffDNA) in the maternal plasma, showing the presence of Y chromosome-specific DNA and RhD in 1997, by …..” Also delete abbreviation (abbreviation is in line 69).
--lines 69-70: “ ….from maternal blood attracted high attention. It became possible later after the introduction of next-generation sequencing (NGS) technologies in 2011 [15]. Merge the two sentences.
--line 86 “ …….. different kind of RNAs, ….” Various RNAs.
--line 87-90: Please modify the sentence.
--line 113: ssDNA is abbreviated in line 85
--line 135: “These are double-stranded DNA fragments with ….” Use abreviation
--line 148; “It was shown that DNA/lipid complexes increased DNA half-life in plasma of mice, suggesting that lipids protect DNA from degradation [32]”. Pleas correct ……. lipids increase DNA half-life in plasma of mice, suggesting that lipids protect DNA from degradation
--line 149: “ ….formation of DNA/lipid complexes ….” DNA-lipid ….
--line 151: Please modify sentence start with Virtosome. Virtosomes are DNA-RNA-lipoprotein complexes!
--line 169: EMV spell out
--line 170: “ ………… damaged metabolism …..” Please modify
--line 172 “ ……….. and induce resistance to therapy …” chemo- radio-therapy?
--line 284 “They proved to act as miRNA ….. “ They shown to
--line 289: SRY please spell out
--line 298: PIWI Spell out
--line 300: Please clarify. Argonaute proteins are highly specialized small-RNA-binding modules and are considered to be the key components of RNA-silencing pathways.
Author Response
All the suggested minor corrections were made in the main text that were indicated with yellow highlight: Line 17, 39, 46, 64-65, 70, 72, 90, 91-93, 144, 156-157, 158, 160, 177, 180, 179, 181, 320, 325, 336, 338
Thank you for your time and effort.
Reviewer 2 Report
The present review by Melinda Szilagyi et al. entitled « Circulating cell-free nucleic acids : main characteristic and clinical applications » deals with the description of the main types of nucleic acids that are present in body fluid especially in plasma. For each type of nucleic acids, the authors gave a short overview on the possibility to use it as a tool in clinical applications either in diagnostic, prognosis, predispoistion to disease, treatment response…
Such type of review has been already published by different labs over the world. Especially several authors from the present submitted review have already published a similar review some years ago (cited as ref 5). Hence it is strongly expected that the present review brings more informations on the state of the art on this very important and promising field of research.
Major points :
1-Because the authors have chosen to review each type of nucleic acids present in body fluid, the potential of such nucleic acids for clinical applications is not reported with sufficient precison and several important data are lacking. In the chapter on cfmiRNA, the reader is balanced from the possibility to the impossibility to use such nucleic acids for therapeutic application. It is recommended that the authors indicate extensively the arguments that allow the reader to know the practical issue that limits their use in clinical practice (either diagnostic value, prognosis and predispoition to diseases). This will give the possibility for the authors to remove inappropriate general sentences from several chapters concluding that there are limitations to use nucleic acids for clinical applications.
2-In the chapter on cfDNA, the authors cited ref 26 as a major paper in the field as the authors described data from lines 113 to 119. This work was published in non specialized journal of moderate impact. It is recommended to add more significant and recent references to support the main conclusions in the present chapter and to extend addition of data to each chapter.
3-Because clinical applications depends mainly on the technology used for mutation detection, it is of importance that the authors indicate limitations (method errors, cost of material, specifically educated personnel and technology); adding a table is recommended in order to compare modern techniques used for mutations detection of cfDNA, ctDNA…… Data on aberrant cfDNA methylation analyses as a diagnostic and prognostic tool have not been indicated in the present review and hence it is recommended to indicate their importance notably in the field of cancer (for instance breast cancer).
4-A Chapter on cfDNA as a robust molecular diagnostic tool for identification of microbes in body fluids is lacking ; it is recommended that the authors add a chapter reviewing the applications of nucleic acids body fluids for fighting bacterial infection such as sepsis.
5-There is only one Figure (Figure 1) in this review that is not of great interest ; it is recommended to add at least one more Figure as suggested below
Minor points :
-Table 2 : genomic DNA/structure : why single-stranded is indicated ?
-Table 3 : needs to be replaced by a Figure
- Line 201 : the sentence is unclear and needs to be modified « However, less information is available about their function »
-Line 207 : « small (18-23nt) molecules » : is it single ou double stranded ?
-Line 208-220 : out of the scope of this review to detail the genesis of miRNA ; it is highly recommended to remove the corresponding sentences.
-Line 227 : the authors have oversimplified the view on the functional and biological roles of miRNA using only 5 references without any more details (54, 55, 56, 64, 65). The authors have to modify in depth this chapter to avoid mis/overinterpretations of data from the litterature.
Author Response
Dear Reviewer,
Thank you for your comments and suggestions. We hope the quality of the ms was increased extemly with the recommended changes.
Thank you for your time and suggestions.
Reviewer 2
1) Because the authors have chosen to review each type of nucleic acids present in body fluid, the potential of such nucleic acids for clinical applications is not reported with sufficient precison and several important data are lacking. In the chapter on cfmiRNA, the reader is balanced from the possibility to the impossibility to use such nucleic acids for therapeutic application. It is recommended that the authors indicate extensively the arguments that allow the reader to know the practical issue that limits their use in clinical practice (either diagnostic value, prognosis and predispoition to diseases). This will give the possibility for the authors to remove inappropriate general sentences from several chapters concluding that there are limitations to use nucleic acids for clinical applications.
The potential clinical application of cfNAs was discussed in more details in the appropriate sections: line 115-127, 277-286. The chapter about cfmiRNAs and the current perspectives on the role of cfNAs as biomarkers were modified: line 248-286, 449-467. Furthermore, a new chapter and table were added about detection methods where the technical possibilities and limitations were detailed: line 401-444, Table 3.
2) In the chapter on cfDNA, the authors cited ref 26 as a major paper in the field as the authors described data from lines 113 to 119. This work was published in non specialized journal of moderate impact. It is recommended to add more significant and recent references to support the main conclusions in the present chapter and to extend addition of data to each chapter.
The reference 26 and the referred section was deleted. The clinical application of cfDNA fragments is referred to a more appropriate journal (Cristiano et al. 2019), ref 26. The clinical application of cfDNA was more detailed: line 115-127.
3) Because clinical applications depends mainly on the technology used for mutation detection, it is of importance that the authors indicate limitations (method errors, cost of material, specifically educated personnel and technology); adding a table is recommended in order to compare modern techniques used for mutations detection of cfDNA, ctDNA…… Data on aberrant cfDNA methylation analyses as a diagnostic and prognostic tool have not been indicated in the present review and hence it is recommended to indicate their importance notably in the field of cancer (for instance breast cancer).
A new chapter and table were added about detection methods where the technical possibilities and limitations were detailed: line 401-444, Table 3. The analysis of methylation pattern of cfDNA was also discussed: line 120-124.
4) A Chapter on cfDNA as a robust molecular diagnostic tool for identification of microbes in body fluids is lacking ; it is recommended that the authors add a chapter reviewing the applications of nucleic acids body fluids for fighting bacterial infection such as sepsis.
A new chapter about celf-free microbial and viral DNA was added: line 197-227.
5) There is only one Figure (Figure 1) in this review that is not of great interest; it is recommended to add at least one more Figure.
Figure 1 was updated. Moreover a new figure about non-coding RNA molecules was added that replaced Table 3 (Figure 2).
Minor points :
-Table 2 : genomic DNA/structure : why single-stranded is indicated ?
It was deleted (Table 2).
-Table 3 : needs to be replaced by a Figure
It was replaced to Figure 2.
- Line 201 : the sentence is unclear and needs to be modified « However, less information is available about their function »
It was modified: line 239-240.
-Line 207 : « small (18-23nt) molecules » : is it single ou double stranded ?
It was corrected: line 249.
-Line 208-220 : out of the scope of this review to detail the genesis of miRNA ; it is highly recommended to remove the corresponding sentences.
MiRNA processing was deleted. The section about cf-miRNAs was rewritten: line 249-251.
-Line 227 : the authors have oversimplified the view on the functional and biological roles of miRNA using only 5 references without any more details (54, 55, 56, 64, 65). The authors have to modify in depth this chapter to avoid mis/overinterpretations of data from the litterature.
The section about cf-miRNAs was rewritten: line 277-286.
Round 2
Reviewer 2 Report
The present version of the review by Melinda Szilagyi et al. entitled « Circulating cell-free nucleic acids : main characteristic and clinical applications » has been well revised according to reviewer comments and can be accepted for publication.
Some typo errors:
line 451 the reason :
line 439 "gold standard"